# Earthworm distributions are not driven by measurable soil properties. Do they really indicate soil quality?

**Mark E. Hodson**[1☺], **Ron Corstanjeb**[2☺], **David T. Jones**[3], **Jo Witton**[1], **Victoria J. Burton**[3], **Tom Sloan**[1], **Paul Eggleton**[3]*

**1** Department of Environment and Geography, University of York, York, United Kingdom, **2** School of Water, Energy and Environment, Cranfield University, Cranfield, United Kingdom, **3** Life Sciences Department, Natural History Museum, London, United Kingdom

☺ These authors contributed equally to this work.
* P.Eggleton@nhm.ac.uk

**Data Availability Statement:** All relevant data are within the manuscript and its Supporting Information files.

## Abstract

Abundance and distribution of earthworms in agricultural fields is frequently proposed as a measure of soil quality assuming that observed patterns of abundance are in response to improved or degraded environmental conditions. However, it is not clear that earthworm abundances can be directly related to their edaphic environment, as noted in Darwin's final publication, perhaps limiting or restricting their value as indicators of ecological quality in any given field. We present results from a spatially explicit intensive survey of pastures within United Kingdom farms, looking for the main drivers of earthworm density at a range of scales. When describing spatial variability of both total and ecotype-specific earthworm abundance within any given field, the best predictor was earthworm abundance itself within 20–30 m of the sampling point; there were no consistent environmental correlates with earthworm numbers, suggesting that biological factors (e.g. colonisation rate, competition, predation, parasitism) drive or at least significantly modify earthworm distributions at this spatial level. However, at the national scale, earthworm abundance is well predicted by soil nitrate levels, density, temperature and moisture content, albeit not in a simple linear fashion. This suggests that although land can be managed at the farm scale to promote earthworm abundance and the resulting soil processes that deliver ecosystem services, within a field, earthworm distributions will remain patchy. The use of earthworms as soil quality indicators must therefore be carried out with care, ensuring that sufficient samples are taken within field to take account of variability in earthworm populations that is unrelated to soil chemical and physical properties.

## Introduction

The study of the spatial variation in a species abundance has often resulted in significant insights into their biology and ecology, e.g. [1]. This has not been the case in pasture earthworms, where the reasons for the spatial patterns of species abundance observed within any

**Funding:** This was funded by a NERC-BESS grant [Scaling and thresholds in earthworm abundance and diversity in grassland agricultural systems, NE/K015338/1. ] The funders had no role in study design, data collection and analysis, decision to publish, or preparation of the manuscript. Dr David Jones took a salary from this grant award.

**Competing interests:** The authors have declared that no competing interests exist.

given field still remain unclear (S1 Table). The apparent random distribution of earthworms at the field scale has been the object of study over many years. Charles Darwin's final publication was a detailed study of earthworms [2]. Among other insights he commented that "even on the same field worms are much more frequent in some places than in others, without any visible difference in the nature of the soil", though later in the same section he suggests that soil moisture and compaction were likely important factors. This is important because it is widely agreed that earthworms, as ecosystem engineers, are fundamental in driving a range of soil processes that give rise to many ecosystem services, particularly in agriculture, e.g. [3–5]. Thus, understanding the distribution of earthworms may allow us to manage the services that they provide.

It is because earthworms drive soil processes that contribute to ecosystem services that it is frequently suggested that earthworms are good indicators of a healthy agricultural soil; earthworm counts feature in a number of schemes for assessing the ecological quality of the soil and farming environment, e.g. [6–11]. In field surveys and controlled laboratory experiments earthworms are known to respond to a variety of key environmental drivers such as pH, temperature, soil texture, soil moisture and soil density e.g. [12] and the spatial distribution of earthworms must in some way represent an integrated response to these, to climatic gradients more broadly which, for example combine effects of temperature and rainfall, e.g. [13,14] and other, environmental variables, for example plant species present [15]. This raises the prospect that spatially resolved measures of earthworm diversity and abundance may provide a good approximation both to soil health/quality and also the ecosystem services delivered by them.

A variety of studies investigating controls on the distributions of earthworms in fields have reached similar conclusions to those of Darwin (S1 Table). In studies using variograms, Spatial Analysis by Distance Indices (SADIE) methods and other statistical methods involving correlations and comparisons of sampling sites and their physical and chemical properties, earthworm distribution has been found to be spatially correlated but also patchy in both arable and pasture systems with typical patches of earthworms/spatial dependence of earthworm numbers on the scale of 20–60 m. The evidence from these studies is that these patches are either unrelated to measured soil properties or correlate with particular site specific soil properties with no consistency across the studies. This raises concerns about the robustness of using earthworm abundance as an ecological indicator of soil quality.

The above studies were carried out at the scale of thousands to tens of thousands of square metres within fields/plots. Despite being carried out on different land uses and often in different climatic zones on different continents they all report the patchy distribution of earthworms. At these scales patchiness can be generated in a number of ways: 1) earthworms moving to patches of soil of "good" quality, 2) self-regulation with a limiting resource decreasing in response to increased earthworm density leading to a decrease in survival/fecundity tied to low mobility such that earthworms are unable to migrate sufficiently rapidly to resource-rich areas when limiting resources fall below a critical level [16], 3) reproduction being more rapid than dispersal [17], 4) predation resulting in significant reductions in the prey population that can not be balanced by reproduction before the predator moves to a new prey-rich area [18] 5) complex interactions with parasites which cause a reduction in fecundity or an increase in mortality in their hosts and the relative mobilities of the parasites and hosts [18], and 6) passive dispersal by humans [19] (note here we are not considering the situation where earthworms are invasive species). All but the first of these causes would suggest that earthworms are not necessarily a good indicator of soil quality. At a larger landscape scale, studies suggest that earthworm numbers increase with decreasing management intensity across different land uses (e.g. arable < pasture < woodland, [20]; pasture< young spruce plantation < old beech coppice, [21] and that changes happen relatively rapidly, within three

years [20]. Schmidt and Briones' recent meta-analysis indicates that most studies on arable fields show that a reduction in tillage of arable fields leads to an increase in earthworm abundance [22]. However, studies on the effects of varying levels of intensification within other land uses, such as pasture which is the topic of the current study, are still lacking.

The existing work on controls on earthworm distributions suggest to us that although earthworm distributions are undoubtedly integrators of soil properties and that although earthworms are responsible for soil processes that give rise to ecosystem services, their use as soil quality indicators is still far from straightforward. To address this issue, we conducted a spatially nested survey of earthworm abundances in pasture fields managed at varying levels of intensity across Britain, at farms located across the dominant national climatological gradients (temperature and rainfall). We used our data to test the hypotheses that at a field scale the spatial variation of earthworm abundance is random whilst at a national scale the variation in earthworm abundance is driven by climate-related gradients in temperature and rainfall that will be reflected in soil temperature and soil moisture values.

## Methods

### Site locations

After obtaining permission from the farm owners, we surveyed four fields at each of seven dairy farms across Britain (Fig 1; grid co-ordinates for all sampling points are given in S5 Table). Farms were chosen on the basis of previous contact with dairy farmers who were willing to allow sampling on their pasture fields, the availability of at least four contiguous fields per farm for sampling and to give locations across the UKs north-south temperature and east-west rainfall gradients. Fields ranged in size from 0.97 to 13.34 ha with a mean size of 3.94 ha (full details are given in S2 Table). For each farm and sample location we obtained geology and predominant soil groups from the Soil surveys of England and Wales [23], and of Scotland, and aspect, elevation and slope from a digital elevation map [24]. Within each farm we sampled four fields covering a gradient of land management intensity, from unimproved rough grazing to improved pasture; fields had been in pasture management for at least three years, and the majority for far longer (S2 Table). Farmers were asked to rank the four sampled fields in terms of management intensity from "lowest" to "low to medium" to "medium to high" to "highest" as part of the process of field selection for sampling. We purposely did not define "management intensity" in terms of a metric for the farmers to rank their fields against as one single metric may not be relevant to all farms; instead we relied on the famers expert judgement and experience with our choice of statistical methods (see below) being driven by the need to be able to incorporate such qualitative measures into our analysis. The classes were determined by conversation with the land owner/farmer and are relative measures within each farm but not between farms. Within each field, we characterised the spatial distribution of earthworms by an intensive spatial sampling scheme where we measured earthworm abundances and soil properties.

### Earthworm sampling and identification

Sampling was carried out 7[th] October– 15[th] December, 2013. We chose these dates as earthworm abundance is less likely to be detrimentally affected by seasonal weather conditions in late autumn; in the hotter, drier months (and the coldest months) earthworms either die or aestivate e.g. [25] which would hinder investigating controls on their distribution. Work conducted for several years in the New Forest has shown that this is the optimal period to sample earthworms [25]. During the sampling campaign none of the fields froze and all had active worms throughout the whole of the sampling period. There may have been overnight frosts

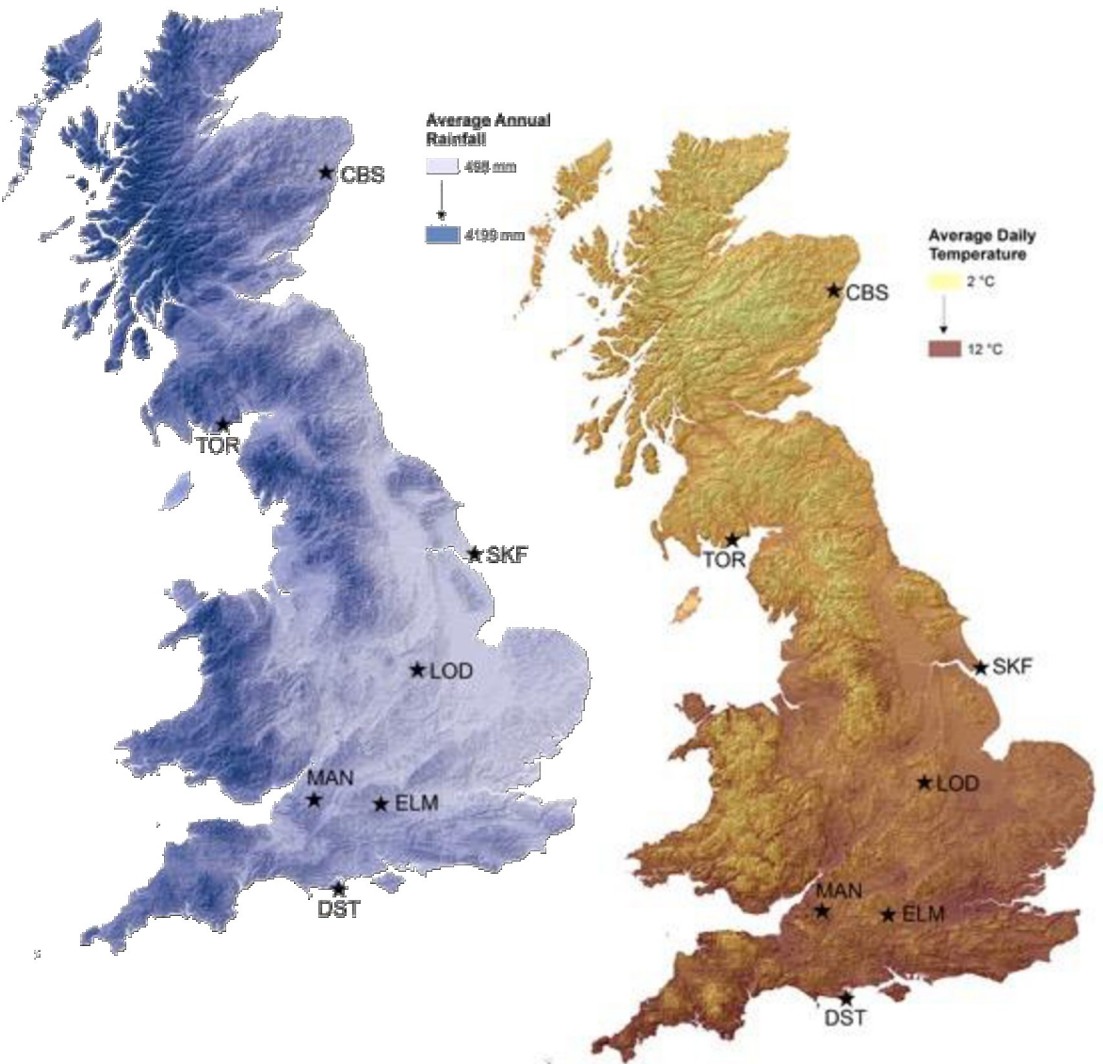

**Fig 1. Sampling locations in the UK and their relationship to average annual rainfall and average annual temperature, illustrating the temperature and soil moisture gradient used to help determine the farms sampled.** Sampling was carried out at SRUC, Craibstone Aberdeen (CBS), Torr Organic, Dumfries (TOR), Skeffling, Yorkshire (SKF), Loddington, Leicester (LOD), Manor Farm, Somerset (MAN), Elm Farm Berkshire (ELM) and Swanage, Dorset (DST). Figure created by Ron Corstanje, University of Cranfield and released under CC By 4.0 License.

towards the end of the sampling period but if there were they did not noticeably affect earthworm activity. We sampled earthworms in pits and preserved them in 80% industrial methylated spirits. Adult individuals (with a clitellum) were identified (by DTJ and VJB) to both species level following [26] and [27] and functional group. Juvenile individuals were grouped into functional group (endogeic, epigeic or anecic species) based on colour, size and external morphology [28]. An average of 18.9 ± 1.4 (std. dev, n = 28) pits were sampled per field with the maximum (and target) number of pits being 20 and the minimum 16. When fewer than 20 pits were sampled this was due to time constraints; typically we had three days sampling time at each farm in an attempt to ensure all the samples collected from a single farm were collected under similar environmental conditions. Within each field the pits were positioned using a random number generator that ensured the distances separating any one pit from every other pit in the same field could vary from a few meters to > 100 m depending on the overall size of

the field. The sampling sites were allocated randomly to ensure no spatial bias, coverage across the field and to ensure that sufficient separation distance was sampled for an effective estimation of the model variogram (see below). Pits falling within 5 m of the field boundary were reassigned a new random position, as field margins are known to have a different invertebrate assemblage from those in the middle of the field [29]. The geolocation of each pit was recorded with a Garmin eTrex 10 GPS. At each pit temperature at 5 cm and 10 cm depth was measured with a soil thermometer and averaged. Four moisture readings were made using a Delta-T Moisture metre (Delta-T Devices, Cambridge, England) with attached Theta probe and then averaged. Each soil pit had a surface area of 20 cm x 20 cm. The earthworms were sampled using the well-established method of hand sorting and mustard extraction, e.g. [30]. The soil monolith (pit) was dug out with a spade to a depth of 10 cm and placed on a plastic sheet. Immediately upon removing the soil 2 litres of mustard solution (12g Colman's mustard powder in 1 L of water) were poured into the soil pit. The extracted soil and plant roots were searched by hand and all earthworms collected. Any earthworms emerging from the bottom of the pit during the 15 minutes following the application of the mustard solution were collected separately. Vegetation cover as % of covered soil was assessed for several broad functional type groups around the vicinity of the pits using the Daubenmire counting method [31].

## Soil sampling and analysis

Two soil samples were collected from each soil pit during the earthworm sampling. One sample was collected from midway down the side of the pit wall using a 95 $cm^3$ density ring and used for bulk density and subsequent measurements on dry soil. The other, c. 50 g, was taken from the homogenised soil that had been excavated from the pit and sorted through for earthworm collection; this soil was used for measurements that required field moist soil. We measured a wide range of explanatory variables from each pit. Field moist subsamples of each soil sample were stored at 4°C and analysed for microbial activity using fluorescein diacetate hydrolysis within one week of sampling [32]. Soil density samples were oven dried and weighed to calculate dry bulk density and soil moisture content. The soil samples were then sieved to < 2mm and analysed for particle size distribution using a Malvern Mastersizer 2000. pH was measured in water using a 10g:25mL ratio [33]. Total carbon and total nitrogen content (%C and %N) of soil was measured by combustion using a Vario MacroElemental (CN) Analyser. Extractable nitrate (nitrate-N) and ammonia (ammonia-N) by 1M KCl and extractable phosphate (phosphate-P) by 0.5M $NaHCO_3$ were analysed by autoanalyser [34]. Bulk composition was determined by aqua regia digest [35] and analysis by inductively coupled plasma–optical emission spectrometry (ICP-OES). Detection limits were calculated for solution analyses by repeated measurements of procedural blanks and precision (which was generally > 95%) by repeated measurements of selected solutions (S3 and S4 Tables). For the aqua regia digests recoveries were > 95% for analysis of the San Joaqiun NIST SRM 2709a.

## Statistical analysis

Our approach comprises an initial set of national (across all farms) and farm level models using Bayesian Belief Networks [36] to determine which are the key factors in determining the total earthworm abundance. Whereas we used Bayesian methods when we were dealing with complex mixtures of expert knowledge, categorical and continuous data we subsequently modelled the species responses to these key factors using generalised linear models as species response curves are more straightforward and can be analysed using standard approaches. At the farm scale, across fields, linear models of coregionalization (LMCR [37]) were then used to describe both the spatial variation in total abundances and their possible spatial co-

dependency to the measured soil properties at the field to farm scale. Here we report the results for all earthworms. We also carried out equivalent analysis for individual earthworm ecotypes (anecic, endogeic and epigeic); these results are presented in the Supplementary information rather than the main text as our key results and conclusions were the same as for the total data set.

Bayesian Belief Networks (BBN) are a set of probabilistic influence networks that are increasingly being used in ecological studies, as they are well-suited to complex data sets and incomplete knowledge, which are common to ecological systems, [36,38]. The BBN modelling was conducted in Netica [39]. The influence network for the BBN included data measured at each sample pit location during field work (soil properties, vegetation cover), field management intensity as described by the local farmers/landowners, and landscape attributes (soil type, geology, aspect and slope) obtained as stated above from the LandIS database (Soil Survey of England and Wales - http://www.landis.org.uk) and from a digital terrain model (represented in Fig 1), and resulting in a range of landscape variables (Elevation, Aspect, Slope, Elev, Soil association, soil unit, soil description, dominant geology, dominate soil series in the association, associated soil series in the associate, historical crop/land-use) in LandIs with the total earthworm abundances as the response variable. Conditional probabilities describe the relationships between the sample pit location, farm and landscape level factors and the earthworm abundances. These were obtained through processing individual cases, where each case represented a point observation of abundance and factor values found at a given location. All nodes in the BBN models were assigned five states, based on round values and distributions for predictors, and equal intervals for the abundance predictions. The conditional probabilities produced for the five levels of abundance were visualised as a heat map using R version 3.2.2 [40]. The resultant BBN structure is shown in the Supporting Information (S1 Fig).

For the species response models we undertook curve fitting for individual earthworm species using generalized linear modelling on the entire dataset (using CANOCO 5 [41], which has a generalized linear model curve fitting module). We used the count number for each species in each monolith as the dependent variable and the chemical and soil variables from the same monolith as the independent variables. We fitted null, linear and quadratic models and calculated AIC and F-values for each model for each species and chose the model in each case that had the lowest AIC and highest F. A Poisson error distribution was used for each species-level model with a log link function, as was appropriate for count data. These were not mixed model as we were interested in patterns across the whole dataset, and we did not use or quote the inevitably over-inflated P-values.

**Linear models of coregionalisation.** The sampling design at the farm level was such that spatial variation of earthworm abundance can be described using variograms, and its spatial covariation with likely drivers by cross variograms [42]. These are models which describe spatial patterns, are used for predictions at unsampled locations (generating maps), and assess the uncertainty associated with these predictions. The variogram represents a measure of dissimilarity as a function of distance between points, and the cross variogram a measure of the joint variability between the earthworm abundances and a particular soil property. From the variogram model we can obtain a set of parameters. The nugget variance ($c_0$), is the point at which the model crosses the intercept and represents the variation observed in the data that is either not spatially correlated or is correlated but at a finer distance than that sampled in this study (5 m). The sill variance ($c_1$) is the value of semivariance at which the variogram model levels off and represents the total observed variation in the dataset under consideration. The ratio of nugget to sill variance ($c_0/(c_0 + c_1)$) expresses the fraction of variation which can be ascribed to spatial processes. Here the larger the ratio, the more of the observed variation can be described as spatially autocorrelated. If the variable under observation approximates a spatially random

process, then this ratio would approach 0. All variogram and cross variogram models were fit individually at each farm using the gstat package [43] in R version 3.2.2 [40].

## Results and discussion

### Observed earthworm abundances

Our full data set (grid references for each sampling point, earthworm data, soil data, landscape variables) is provided in the Supporting Information (S5 Table). Across all the sample sites, juvenile earthworms were more abundant than adults of any one species. Of the three ecotypes of earthworm, endogeics were more abundant than epigeics which were in turn more abundant than anecic species (Fig 2A).

There is a paucity of systematic distribution data for earthworm species in the UK [44]. However, our data are typical of other pasture surveys of earthworms in the UK and similar climatic regions. As with other studies populations are dominated by juvenile and particularly juvenile endogeic earthworms e.g. [45–49], while among adult populations, endogeic earthworms, notably, *Allolobophora chlorotica* and *Apporectodea caliginosa* often dominate with *Lumbricus rubellus* and/or *Lumbricus castaneus* being the most abundant epigeic species; *Aporrectodea longa* and *L. terrestris* are the only two UK anecic species e.g. [46–51]. All the farm sites apart from Manor Farm had significant differences in earthworm numbers between fields (ANOVA on Ranks for BS, SKF, LOD, MAN, ELM, DST; ANOVA for TOR; $p < 0.05$) but using this simple linear modelling approach, we found no consistent pattern between relative level of management intensity and earthworm numbers. The relationship between earthworm numbers, management intensity and field properties was therefore further investigated using Bayesian Belief Networks.

### National level drivers of earthworm distribution

At a national scale, the Bayesian belief network developed for the entire data set, was able to describe about 57% of the observed variation in abundances (absolute error of 19.72, and root mean square error of 26.12). Given the scale and nature of the ecological dataset, these results are not unreasonable. For this model, the overall most influential variables were found to be

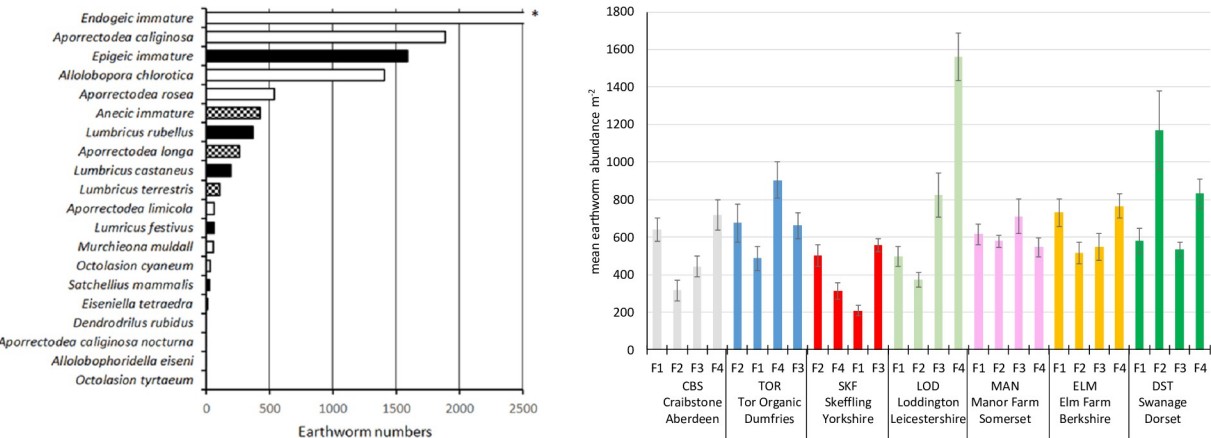

**Fig 2.** Abundances of earthworms by (a) species (total numbers). Total numbers = 13298 earthworms (endogeics = 77%, epigeics = 17%, anecics = 6%).* = continues to 6126 individuals; open bars indicates endogeic species, closed bars epigeics and checked bars anecics and (b) sampling site (mean abundance m$^{-2}$ ± standard error, n = 15–20). F1 to F4 for each sampling site are fields 1 through 4 and for each farm are arranged in order from lowest management intensity on the left to highest intensity on the right. The intensity levels are relative and not necessarily comparable between farms.

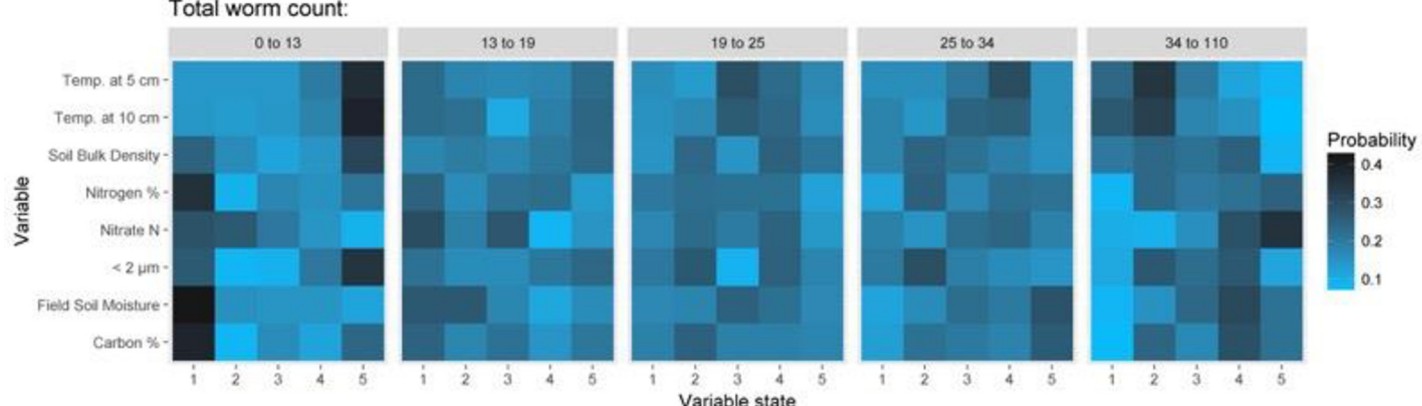

**Fig 3. Panels show the sensitivity of the overall Bayesian belief network to main continuous variables used to predict values of total earthworm counts across all 7 farms.** Values of the continuous variables were split into 5 bins (where Bin 1 has the lowest values and 5 the highest) as defined for this dataset. For details of bin ranges see Table 1; the more intense the colour the more likely this value is for this number of earthworms. For example, at a depth of 5 cm, if temperature is in the range 12–16˚C then earthworm abundance is most likely to be low (0–13 individuals), if the observed temperature is in the range 9.1–10.8˚C, then the earthworm abundance will probably be in the range of 25–34 individuals, and at an observed temperature range 6.9–9.1˚C then the abundance is most likely to be high (34–110 individuals), at a farm scale.

Field temperature, Soil bulk density, Moisture and Nitrate-N followed by Soil type and Underlying geology. The main analytically measured continuous variables were % clay, % carbon, % nitrogen, Soil bulk density, Temperature at 5 and 10 cm and Soil moisture. The sensitivity of the BBN is presented in Fig 3, and Table 1. There are clear climatic effects in the abundance data, with field moisture and temperature as key drivers. Management intensity, as it was conditional on a particular farm, was modelled as a child node of farm, and therefore, at this level, had little influence on abundances. The sensitivity of the BBN at ecotype level (S6 Table) gives similar results as the analysis for the entire data set though interestingly epigeic earthworms were determined to be less sensitive to soil moisture levels than endogeic and anecic earthworms. This difference may reflect the litter dwelling nature of the epigeic earthworms compared to the soil dwelling nature of endogeic and anecic earthworms.

When species level response curves were considered in the full dataset by simply fitting Generalize Linear Models to soil parameters (S7 Table), we found that bulk density and nitrate level in particular are strongly related to earthworm numbers across a range of species (Fig 4). However, a majority of the plots show that a unimodal curve was the best fit response to the environmental gradients. Nitrate-N, for example has narrow optima for *Allobophora chlorotica*, *Aporrectodea caliginosa* and *Aporrectodea longa*, showing that they are less abundant at low and high nitrate concentrations. Bulk density shows a much flatter response for most species while still typically showing optimal values. The response to soil moisture is more idiosyncratic, because of the presence of *Aporrectodea limicola*, a species often found in waterlogged soils [27] and *Aporrectodea caliginosa*, which is tolerant of flooding [52,53]. For bulk density, nitrate-N, pH and soil moisture content *A. chlorotica* show the narrowest peaks, suggesting that this species has more marked preferences, and possibly sensitivities, to these soil properties. Both bulk density and nitrate were highlighted in the national level BBNs as exerting important controls on earthworm numbers.

## Farm level drivers of earthworm abundance

Bayesian Belief Networks produced for the individual farms using both the entire data set (Table 2) and for different earthworm ecotypes (S8 Table) are consistent with the analysis

**Table 1. The ranges of soil properties associated with each of the 5 bins used in the BBN.**

| Total number of earthworms | % clay | % carbon | NO₃-N | % nitrogen | Density | Temperature @ 10 cm | Temperature @ 5 cm | Soil moisture content | State |
|---|---|---|---|---|---|---|---|---|---|
| 0–13 | 0–0.04 | 1.5–3.4 | -0.3–0.5 | 0.15–0.3 | 0.23–0.69 | 4.7–6.9 | 4.6–6.9 | 17–41 | 1 |
| 13–19 | 0.04–0.5 | 3.4–4.6 | 0.5–1.1 | 0.3–0.45 | 0.69–0.83 | 6.9–9.2 | 6.9–9.1 | 41–47 | 2 |
| 19–25 | 0.5–0.7 | 4.6–6.3 | 1.1–2.9 | 0.45–0.56 | 0.83–0.92 | 9.2–10.9 | 9.1–10.8 | 47–54 | 3 |
| 25–34 | 0.7–1.1 | 6.3–9 | 2.9–6 | 0.56–0.8 | 0.92–1.1 | 10.9–11.9 | 10.8–12 | 54–61 | 4 |
| 34–110 | 1.1–7.7 | 9–42 | 6–105 | 0.8–2.2 | 1.1–1.7 | 11.9–20 | 12–16 | 61–96 | 5 |

Bins 1–5 correspond to the positions of the boxes indicated in Fig 3. The net had an absolute error of 19.72, and root mean square error of 26.12.

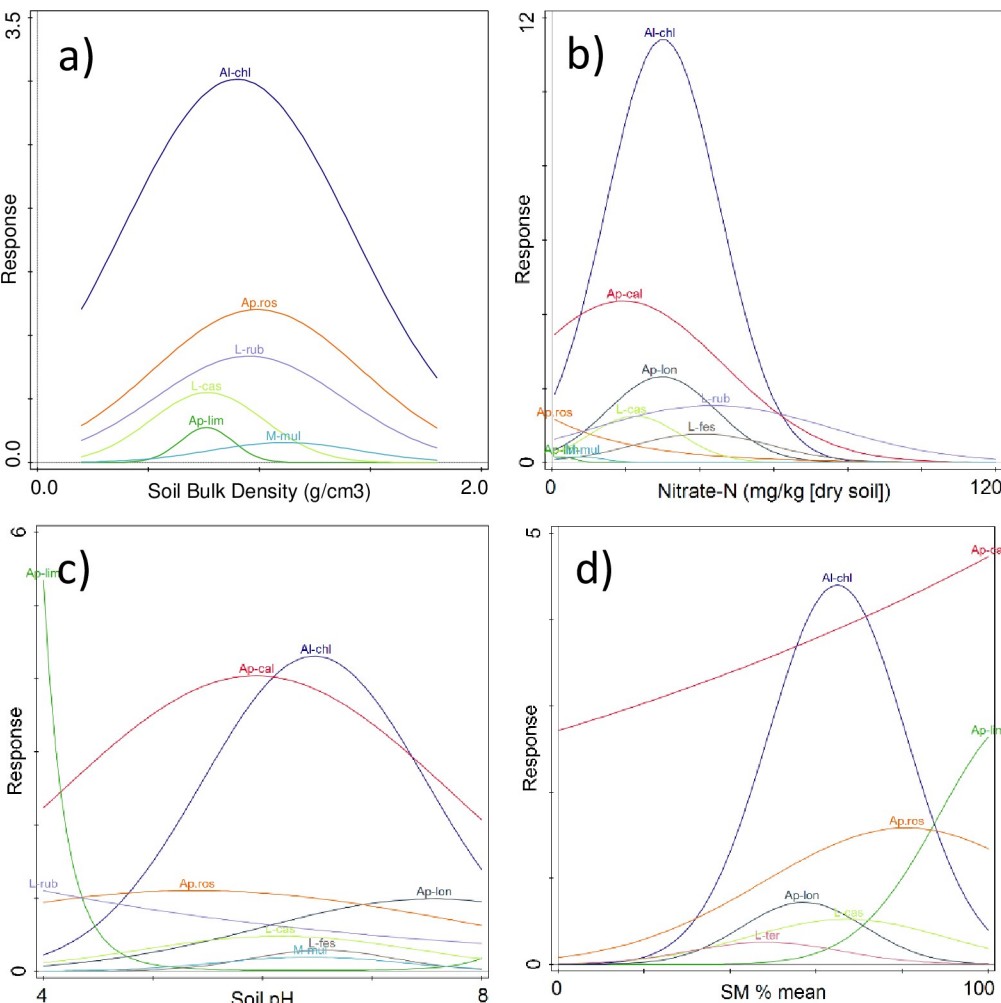

**Fig 4. Generalized linear model fits between measured soil parameters and numbers of earthworms at a species level across all sampling sites.** a) bulk density, b) nitrate-N, c) pH, d) soil moisture content (SM%). Details of the models including uncertainties are given in S7 Table. Only species where numbers vary with the particular soil parameter are shown. (Al-chl = *Allolobophora chlorotica*, A pros = *Aporrectodea rosea*, L-rub = *Lumbricus rubellus*, L-cas = *Lumbricus castaneus*, Ap-lim = *Aporrectodea limicola*, Ap-cal = *Aporrectodea caliginosa*, Ap-lon = *Aporrectodea longa*, M-mul = *Murchieona muldall*, L-fes = *Lumbricus festivus*, L-ter = *Lumbricus terrestris*.

**Table 2. Results from the Bayesian Belief Networks developed for each individual farm.**

| Site | LOD | SKF | DST | TOR | CBS | ELM | MAN |
|---|---|---|---|---|---|---|---|
| Abs Error | 4 | 1.43 | 6.05 | 2.00 | 2.5 | 2.47 | 2.81 |
| RMSE | 6.74 | 2.06 | 12.56 | 3.44 | 4 | 4.50 | 6.13 |
| Top 10 variables | Management intensity | Management intensity | Management intensity | Domsoils | Management intensity | Soil water content | Iron |
| | Potassium | Calcium | FDH activity | Geology | Slope | FDH Activity | Slope |
| | % carbon | Domsoils | Temp at 10 cm depth | Management intensity | Temp at 10 cm depth | % carbon | Soil bulk density |
| | % nitrogen | Geology | Aluminium | Potassium | FDH activity | Management intensity | Manganese |
| | Soil pH | Magnesium | Temp at 5cm depth | $PO_4$-P | C:N | Phosphorus | $NH_4$-N |
| | Phosphorus | Strontium | Lead | Lead | Soil pH | Aluminium | Strontium |
| | Magnesium | Elevation | Soil pH | Elevation | Elevation | Aspect | Calcium |
| | $NO_3$-N | Temp at 5 cm depth | Elevation | Temp @ 5cm | Zinc | Soil bulk density | % nitrogen |
| | Aluminium | Phosphorus | Slope | Sodium | % clay | Iron | % clay |
| | Sodium | Soil water content | Cadmium | Temp at 10 cm depth | Aspect | Field soil moisture | Aluminium |

The top portion describes the performance of the networks as Absolute (Abs) Error and Root Mean Square Error (RMSE) on the earthworm species count estimates. The subsequent portion of the table describes results from a sensitivity analysis on the models by ranking, from top to bottom, the most influential variables in each model.

carried out at the national level. The networks describe earthworm abundance generally well, with an RMSE between 12 and 2 and an absolute error between 6 and 1 counts. In terms of the most significant variables, Management Intensity was consistently identified as the main factor at field scale for the average earthworm abundance within a farm with variables relating to climate (e.g. temperature and soil water content) consistency being included in the identified secondary variables (Table 2).

## Field level spatial patterns of earthworm abundances

We were able to determine spatial autocorrelation, and fit authorised variogram models to 5 of the 7 farms, with DST and MAN exhibiting only pure nugget variation (Fig 5, Table 3). Of these five farms, SKF exhibited the longest range at 114 m, with the others ranging between 19 and 30 m. This suggests that for the majority of the farms (with the exception of SKF), earthworm populations exhibited either short range spatial autocorrelation or no spatial autocorrelation at all. Where autocorrelation was observed, the variance component which was spatially autocorelated was generally substantial (from an estimated 37% to 91% of the observed variation in earthworm abundance). These variogram results are in general agreement with other studies across a range of land uses, climates and species (S1 Table), and it is therefore clear that in any given field, the spatial variability in earthworm abundances will be short range and patchy in nature, i.e. earthworm abundance will be high in patches of c. 30–100 m width, and those patches appear to be randomly distributed within fields. Variograms for the different ecotypes are presented in the Supplementary information (S9 Table) and show similar results for each ecotype.

For those farms where we observed spatial autocorrelation in earthworm abundances, we subsequently fitted LMCRs but observed no spatial covariation with the earthworm abundances in any of the soil properties (e.g. soil organic carbon, pH, density, nitrate-N) at the distance intervals for which earthworms show spatial autocorrelation. This would indicate that

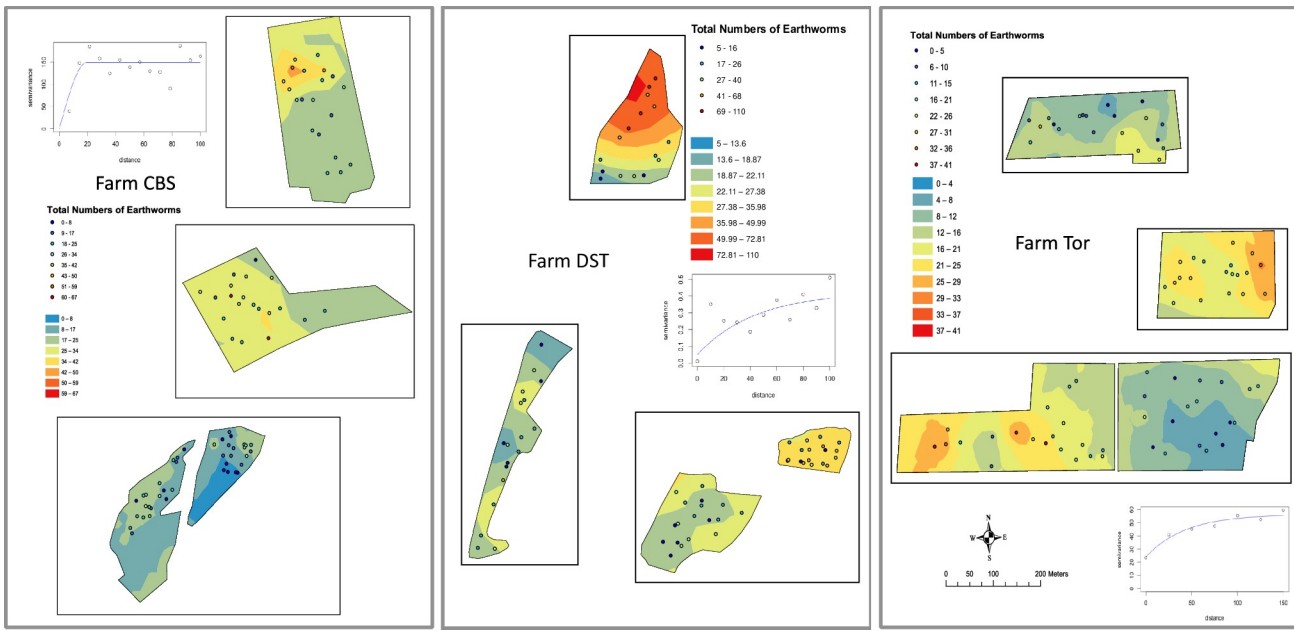

**Fig 5.** Within field and farm distribution of earthworms for each of the four fields at three representative sites, a) CBS, b) DST and c) SKF. Contoured colours represent the interpolated (predicted) values of total earthworm counts, coloured dots represent the observed counts within each sampling point. Inserts show the variograms used to generate the interpolated surfaces, with the separation distances on the x-axis, and semivariance (earthworm counts, log for DST) on the y-axis.

none of these properties were key drivers of the abundance of the earthworms at the farm scale despite our species response curves (Fig 4) and the results of the full dataset BBNs (Fig 3, Tables 1 and 2) and the well-established literature concerning the physico-chemical factors that earthworms are sensitive to (see the many references in [12]). Since our results show that soil properties of themselves do not control the spatial variability of earthworm communities they suggest that instead biological and ecological factors, such as rates of reproduction and migration, anthropogenic interventions, predation and parasitism, as proposed by Satchell [17], may be the driving factors for short scale variation in abundance in a way that then confounds the short range effect of edaphic factors. We cannot be certain which of those biotic factors were the most important as the sampling protocol did not allow us to examine local biotic factors in sufficient detail.

**Table 3. Variogram parameters for the seven sites.**

| Site | Range (m) | $c_i$ | $c_o$ | $c_i/(c_i+c_o)$ |
|---|---|---|---|---|
| LOD | 27 | 230 | 22 | 91% |
| SKF | 114 | 33..7 | 22 | 61% |
| DST | - | - | - | - |
| TOR | 23 | 47 | 80 | 37% |
| CBS | 19 | 124 | 20 | 86% |
| ELM | 30 | 103 | 40 | 72% |
| MAN | - | - | - | - |

Range reflects the extent to which the spatial autocorrelation extends, $c_i$ is the sill variance, and $c_o$ is the nugget variance, $c_i/(c_i+c_o)$ represents the variance component that is spatially autocorrelated. No spatial autocorrelation was found for DST and MAN.

## Implications of our results for the use of earthworms as indicators of soil health and soil management

For most fields, earthworms are spatially correlated at short distances and a knowledge of their distribution does not provide spatially resolved information about factors such as soil pH, density, nitrate content and organic matter content. Thus, individual samples are unreliable indicators of earthworm populations and other soil properties. Further, the level of variability in earthworm numbers between sampling sites within a field suggests that for any monitoring scheme that requires a reliable estimate of earthworm numbers, whether earthworm numbers are of direct interest or are being used as a proxy for other specific soil properties such as organic matter content or more diffuse measures such as soil health/quality, multiple samples are required because these properties show significant within-field variation. The schemes detailed in [7] and [8] suggest sampling 3 or 4 sites per field to assess soil quality with earthworm numbers as one of multiple tests proposed on those samples. A recent national level survey aimed at using earthworms to assess soil health of arable fields used a sampling density of 10 pits per field with a total survey time of 60 minutes per field [11]. The survey achieved high levels of participation and cost-effectiveness and supported the accepted wisdom that tillage has a negative effect on earthworm numbers. It was suggested that numbers of pits could be reduced to 5 per field with little decrease in the quality of the results. In our study, if we assume comparable threshold qualitative values of > 66% of pits containing 16 earthworms or more as described by Stroud [11], then in order to determine the difference between good and poor quality soil in pasture fields on the basis of earthworm abundances, based on a power analysis of our data, we would recommend about 8–10 pits per field.

Much interest in earthworm populations stems from their role in soil processes that deliver ecosystem services [3–5]. While it is well established that moving from conventional to minimum- or no-till management can lead to increases in earthworm numbers in arable fields [22] our results suggest that pasture fields can also be managed to enhance earthworm numbers. At a national scale, earthworm abundance appears to be related to both soil temperature and moisture reflecting natural gradients but also soil nitrate levels and soil density. In broad terms at the national level more intensely managed farms will have higher N inputs leading to higher levels of nitrate N e.g. [54–57]. Similarly, increasing intensity of farm management is likely to be linked to more on-field traffic with consequent compaction and density increases. Our definition of management intensity at the farm level is qualitative and based on discussion with the resident farmers. Thus, our ranking of relative intensity is not directly comparable between farms. However, our BBN networks for individual farms (Table 2), the lack of a linear relationship between earthworm numbers and intensity at each farm (Fig 2B) and the unimodal species response curves for soil density and soil nitrate-N (Fig 4) suggests that there is a role for management interventions to maximise earthworm populations, for example through fertiliser or organic matter amendments, to optimise earthworm-friendly conditions.

## Conclusions

Although Darwin's insights were based on sparse quantitative data it seems that his essential intuition was correct: at the field scale, the spatial variation in earthworm abundance is controlled or at least heavily influenced by some factor other than soil properties, most likely biotic (possibly reproduction, migration rates, competition and predation), giving the appearance of a spatially random distribution of ~30 - ~100 m patches of high earthworm abundance. This suggests that the use of individual based models, that incorporate biological behaviour e.g. [58–61] are likely to be a better route for understanding the distribution of earthworms in fields than measures of soil properties.

However, at a national level earthworm distributions are clearly influenced by both natural gradients related to temperature and moisture and (in a non-linear fashion) by factors such as soil nitrate and density that will be influenced by levels of intensity of farm management. Thus, whilst it is generally accepted that reducing fertiliser use and minimising compaction are important interventions to increase the sustainability of arable farming such interventions will impact on earthworm populations in a non-linear fashion. This suggests that management interventions are possible to increase earthworm numbers, and thus enhance the ecosystem services that their actions in soil give rise to, at the farm scale. This means examining the farm as a whole, taking into account its previous land use and cropping history, and not assuming that a single field can be managed alone to reach the farmer's expectations.

## Supporting information

**S1 Fig. Typical BBN structure for the entire dataset.**
(DOCX)

**S1 Table. Summary of selected previous studies that investigated the spatial distribution of earthworms.**
(DOCX)

**S2 Table. Land use history of sampling sites according to land owners.** Numbers indicate minimum time that field has been used as pasture. "Never" indicates no use other than pasture.
(DOCX)

**S3 Table. Detection limits for soil analyses.**
(DOCX)

**S4 Table. Repeat ICP-OES analyses for precision.**
(DOCX)

**S5 Table. Earthworm and soil data set.**
(DOCX)

**S6 Table.** a. Sensitivity of BBN for epigeic earthworms. See Fig 3 and Table 1 of main paper for how to interpret these tables. b. Sensitivity of BBN for endogeic earthworms. See Fig 3 and Table 1 of main paper for how to interpret these tables. c. Sensitivity of BBN for anecic earthworms. See Fig 3 and Table 1 of main paper for how to interpret these tables.
(DOCX)

**S7 Table. Summary of general linear model fits to species data–separate Excel file.**
(XLSX)

**S8 Table.** a. Results from the Bayesian Belief Networks developed for each individual farm for epigeic earthworms. See Table 2, main manuscript, for further information. b. Results from the Bayesian Belief Networks developed for each individual farm for endogeic earthworms. (See Table 2, main manuscript, for further information). c. Results from the Bayesian Belief Networks developed for each individual farm for anecic earthworms. See Table 2, main manuscript, for further information.
(DOCX)

**S9 Table.** a. Variogram parameters for the seven sites for epigeic earthworms. See Table 3, main manuscript for details. b. Variogram parameters for the seven sites for endogeic earthworms. See Table 3, main manuscript for details. c. Variogram parameters for the seven sites

for anecic earthworms. See Table 3, main manuscript for details.
(DOCX)

## Acknowledgments

We thank Cecilia Soderholm, Georgiana Datcu, Hannah Measey, Irfaan Junaideen, Joshua March, Linda Beer, Michelle Sutton, Natalie Kay, Bobby Day, Salma Mustafa, Sholto Holdsworth, Sonya Hallett, Veronika Moore, Yamini Panchaksharam, Fevziye Hasan, Jo Smith and Kelly Inward for their help with field work and the land owners for access and Phil Platts for providing an internal review of the paper.

## Author Contributions

**Conceptualization:** Mark E. Hodson, Ron Corstanjeb, Paul Eggleton.

**Data curation:** Ron Corstanjeb, Paul Eggleton.

**Formal analysis:** Mark E. Hodson, Ron Corstanjeb, Paul Eggleton.

**Funding acquisition:** Mark E. Hodson, Paul Eggleton.

**Investigation:** Mark E. Hodson, Ron Corstanjeb, David T. Jones, Jo Witton, Victoria J. Burton, Tom Sloan, Paul Eggleton.

**Methodology:** Paul Eggleton.

**Project administration:** Mark E. Hodson, Paul Eggleton.

**Resources:** David T. Jones, Paul Eggleton.

**Software:** Ron Corstanjeb.

**Supervision:** Mark E. Hodson.

**Validation:** Mark E. Hodson, David T. Jones, Jo Witton, Victoria J. Burton, Tom Sloan.

**Visualization:** Ron Corstanjeb, David T. Jones, Paul Eggleton.

**Writing – original draft:** Mark E. Hodson.

**Writing – review & editing:** Mark E. Hodson, Ron Corstanjeb, David T. Jones, Jo Witton, Victoria J. Burton, Tom Sloan, Paul Eggleton.

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
