## [Decision Letter · Decision Letter 0]

12 Jan 2021

PONE-D-20-33339

Darwin was right, in any given field, the spatial variability of earthworm communities in pastures isn’t driven by measurable soil properties.

PLOS ONE

Dear Dr. Eggleton,

Thank you for submitting your manuscript to PLOS ONE. After careful consideration, we feel that it has merit but does not fully meet PLOS ONE’s publication criteria as it currently stands. Therefore, we invite you to submit a revised version of the manuscript that addresses the points raised during the review process.

The reviewers have made some helpful suggestions to improve clarity in parts of the manuscript.

We look forward to receiving your revised manuscript.

Kind regards,

Manu E. Saunders

Academic Editor

PLOS ONE

Journal Requirements:

2. In your Methods section, please provide additional location information of the study sites, including geographic coordinates for the data set if available.

"The funders had no role in study design, data collection and analysis, decision to publish, or preparation of the manuscript"

5. We note that Figure 1 in your submission contain map images which may be copyrighted. All PLOS content is published under the Creative Commons Attribution License (CC BY 4.0), which means that the manuscript, images, and Supporting Information files will be freely available online, and any third party is permitted to access, download, copy, distribute, and use these materials in any way, even commercially, with proper attribution. For these reasons, we cannot publish previously copyrighted maps or satellite images created using proprietary data, such as Google software (Google Maps, Street View, and Earth). For more information, see our copyright guidelines: http://journals.plos.org/plosone/s/licenses-and-copyright.

5.1.    You may seek permission from the original copyright holder of Figure 1 to publish the content specifically under the CC BY 4.0 license. 

5.2.    If you are unable to obtain permission from the original copyright holder to publish these figures under the CC BY 4.0 license or if the copyright holder’s requirements are incompatible with the CC BY 4.0 license, please either i) remove the figure or ii) supply a replacement figure that complies with the CC BY 4.0 license. Please check copyright information on all replacement figures and update the figure caption with source information. If applicable, please specify in the figure caption text when a figure is similar but not identical to the original image and is therefore for illustrative purposes only.

6. Please include a copy of Table 2 which you refer to in your text on page 15.

Reviewers' comments:

Reviewer's Responses to Questions

**Comments to the Author**

1. Is the manuscript technically sound, and do the data support the conclusions?

Reviewer #1: Yes

Reviewer #2: Yes

2. Has the statistical analysis been performed appropriately and rigorously? 

Reviewer #1: Yes

Reviewer #2: Yes

3. Have the authors made all data underlying the findings in their manuscript fully available?

Reviewer #1: Yes

Reviewer #2: Yes

4. Is the manuscript presented in an intelligible fashion and written in standard English?

Reviewer #1: Yes

Reviewer #2: Yes

5. Review Comments to the Author

Reviewer #1: I enjoyed reading the manuscript about an interesting topic in earthworm ecology that has been debated since Darwin, who stated that earthworms within a field are patchy distributed without any clear visible differences in the characteristics of the soil.

With a well-designed survey at seven different farms in the UK along an environmental gradient, the authors indeed showed that earthworm density at field scale (within field) was determined by abundance itself and not by any edaphic conditions. At a national scale (between fields and farms), however, earthworm numbers are predicted by density, nitrate levels in the soil, soil temperature and soil moisture. The authors conclude that the patchiness within a field is probably determined by biological factors such as dispersal rate and predation.

This is not new. As the authors also mention in the Introduction (with a list in the Supplementary information), other studies already showed that Darwin was right. Earthworms are indeed patchy distributed without a clear soil characteristics that determines the pattern. However, the authors try to argue that this has consequences for using earthworms as a soil quality indicator. Because of their important role as ‘ecosystem engineers’ and their positive effects on the soil, earthworms are often used as indicators for ecological soil quality, but this is still far from straightforward as they showed with this study. I think this should be highlighted more in this manuscript.

The structure of the manuscript should therefore be improved. In the analysis and interpretation of the results, the authors should make a clear distinction between national scale, farm scale and field scale. The same is true for total earthworm approach versus the species approach. By doing so, it makes it more easier to interpret the consequences of different sampling designs for earthworms as bio-indicators. Furthermore, the authors should then elaborate more on the consequences of these results in the light of agroecology and how the ecosystems services earthworms provide can be promoted in farmland.

I also would recommend the authors to analyse the data based on the ecological group data instead of the species data only, as most studies about earthworms as bio-indicators use the classification epgigeic, endogeic and anecic (e.g. the national survey which is discussed in the Conclusion). It will probably also increase the power as all earthworms (including the juveniles) are included.

Based on these issues I would recommend major revisions of this manuscript.

Minor issues:

Title: Based on the previous suggestions, I would change the title as other studies already proved that Darwin was right. If the main message is that earthworm abundance is not a good indicator for soil quality, make this clear in the title.

Abstract:

Lines 44-45: From the main text it is not clear how species abundance can be effectively used as ecological indicators.

Lines 45-48: Indicator species are not mentioned in the main text, Remove, repetition of what has been said before.

Introduction:

Lines 55-57: Please add references

Lines 57-58: Please add references

Lines 59-61: Although Darwin also ends the paragraph where he mention the patchiness of earthworms, by stating that it is most likely that soil moisture and compaction are important factors determining the distribution of earthworms…

Lines 78-79: Please add references, or refer to Table S1 here.

Lines 90-91: Based on these numbers it is most likely that these studies were carried out at field scales, please mention that.

Line 91: “…and often in different climatic zones on different continents.” Than in your study? Why mention this?

Line 93: Remove the sentence between brackets as you also mention this after this list.

Lines 91-98: Point 1 refers to heterogeneity of the soil and as a result earthworms move to “good” patches (cause-effect). Please elaborate more how predation and parasitism could generate patchiness, and preferably ad references.

Lines 105-106: What other land uses do you mean and why would that be relevant for this manuscript?

Lines 114-116: Based on the information in the Introduction, I would hypothesize that the variation in earthworm abundance at the national scale is driven by management intensity and not climate as this is not discussed. Please provide information to the Introduction about climate to justify the hypothesis.

Methods:

Line 120: All dairy farms? Please mention this

Line 122: Which soil properties were used to select the fields? Soil type?

Lines 125-128: What were the average size and age of the fields? Perhaps you can include this information in Table S2.

Lines 128-130: What was the management intensity based on? The amount of fertilizer/manure application? Soil disturbance? Traffic intensity? Cattle density? From the Introduction I assume that it is disturbance, but how does this look like on a farm? Were the field with highest intensity the youngest?

As this classification is qualitative and based on discussion with the farmers, perhaps you could make it more quantitative by using nitrate levels or bulk density as a measure of management intensity?

Line 130-131: Was this done before or after the selection of the fields?

Lines 143-146: What about freezing conditions during fieldwork? Earthworms will also die or aestivate under freezing conditions which were more likely to occur during the fieldwork period.

Lines 150-152: How were the number of samples per field determined?

Lines 160-161: Why?

Lines 172-173: What is the rationale behind measuring vegetation cover? The unit is % covered soil?

Lines 179-180: Already mentioned in lines 161-163

Lines 207-210: Which soil properties were included? How were field management and landscape attributes included (units)?

Line 219-221: How was the response variable entered? What were the explanatory variables? Did you use a general linear model (as mentioned in the results section) or a generalized linear model (as mentioned here)? If so, what was the random factor?

Results and Discussion:

Lines 297-298: With temp 5 cm most important and %carbon least important?

Lines 348-352: The main conclusion is that the distribution within fields is likely to be determined by biotic conditions, however, the argumentation for this is very thin, please elaborate more on this and add more references to your conclusions. How could reproduction rate explain the patterns? Or dispersal? Or predation? And is this equal among species or ecological groups?

Conclusions:

Lines 388-391: What were the threshold qualitative vales? And were these also for pasture land? As Stroud also looked in arable land. And what about other studies that use earthworms as indicators?

Lines 405-407: It would be nice if you could argue a bit further how this would look like.

Reviewer #2: This study is an interesting and statistically rigorous examination of spatial variability in earthworm communities across multiple scales. Similar studies have generally looked at a single field in detail or have looked at more sites but only very coarsely, and thus I think it is a very useful contribution to knowledge in this area. I have a number of minor comments:

Line 68 – specify that it is healthy “agricultural” soil that they are suggested to be good indicators of

Table S1 – since there are other studies that have examined earthworm distributions in forests that are not mentioned at all, it would be good to include in the title that these are studies that investigated the spatial distribution of earthworms in “agricultural/savannah” habitats. (I agree it makes sense to only list studies in these open habitats though)

In line 92 - 98, there is a list of how patchiness in earthworm populations can be generated. What about cases where earthworms are spreading in a field for the first time either because disturbance removed them from the area, or because they are invading? Then introduction by humans or passive dispersal by humans or natural vectors could also lead to patchiness (see discussion in Eijsackers 2011 Applied Soil Ecology). These potential sources of patchiness do not seem like they are quite encompassed in any of the items in the list.

Line 114-116 – There is some discussion about how management intensity affects earthworm abundance earlier in the introduction, but not much mention of climate (only in a list with other drivers on line 72), but then the hypothesis here is that climate will drive earthworm abundance at a national scale. I think this is a reasonable hypothesis but it doesn’t seem to follow clearly from the introduction – can you add some explanation (could cite another paper or two (e.g., Phillips et al. 2019 Science)) to provide more background for this hypothesis?

Line 155 – Somehow it is still slightly hard to picture how the pits were positioned (was the field divided into a grid and then the random numbers were used on that, or something?). Please clarify this explanation if possible.

Line 176 For the soil sampling, what depth was the soil collected from? It sounds like it was just collected from the bottom of the soil pit, but was it actually a 10 cm depth core? What volume of soil was collected for the bulk density measurement?

Statistical analysis – I haven’t ever run a BBN model but that analysis seems well-explained. Why did you switch over to AIC/p-values for species response models, versus continuing with a Bayesian approach?

Although the explanation of the variograms/cross variograms is pretty standard I think, could you add a reference here?

Line 390 – can you provide any details on how the power analysis was done?

Figure 5 – this figure seems somewhat disorganized e.g., is the panel that is between a and b part of a) or b)? Placing the variograms in the same position within each panel and adding labels to some of the maps might help. The file names of the layers such as “[DST.txt_Features].[TOTall]” could be deleted or changed so that they are more informative for readers as well.

6. PLOS authors have the option to publish the peer review history of their article (what does this mean?). If published, this will include your full peer review and any attached files.

Reviewer #1: No

Reviewer #2: No

---

## [Author Response · Author response to Decision Letter 0]

22 Jul 2021

Response to reviewers

Below we detail our responses to the reviewers in red italics. Line numbers in the reviewers comments refer to original line numbers, line numbers in our response refer to the revised manuscript without track changes enabled.

We have checked and edited where needed.

2. In your Methods section, please provide additional location information of the study sites, including geographic coordinates for the data set if available.

National grid co-ordinates are presented for each sampling point in the SI information (Table S5). We now make this point in the main text at line 123. Table S2, which is already referenced in this section includes land use history data.

We now state clearly at the start of our “Site locations” section (Line 122) that we had permission from the farm owners to carry out our sampling 

"The funders had no role in study design, data collection and analysis, decision to publish, or preparation of the manuscript"

a. Please clarify the sources of funding (financial or material support) for your study. List the grants or organizations that supported your study, including funding received from your institution.

d. If you did not receive any funding for this study, please state: “The authors received no specific funding for this work.”

This is now included in the cover letter.

5. We note that Figure 1 in your submission contain map images which may be copyrighted. All PLOS content is published under the Creative Commons Attribution License (CC BY 4.0), which means that the manuscript, images, and Supporting Information files will be freely available online, and any third party is permitted to access, download, copy, distribute, and use these materials in any way, even commercially, with proper attribution. For these reasons, we cannot publish previously copyrighted maps or satellite images created using proprietary data, such as Google software (Google Maps, Street View, and Earth). For more information, see our copyright guidelines: http://journals.plos.org/plosone/s/licenses-and-copyright.

5.1. You may seek permission from the original copyright holder of Figure 1 to publish the content specifically under the CC BY 4.0 license. 

We now clearly state in the figure caption that this image was created for the paper and is released under CC By 4.0 License.

5.2. If you are unable to obtain permission from the original copyright holder to publish these figures under the CC BY 4.0 license or if the copyright holder’s requirements are incompatible with the CC BY 4.0 license, please either i) remove the figure or ii) supply a replacement figure that complies with the CC BY 4.0 license. Please check copyright information on all replacement figures and update the figure caption with source information. If applicable, please specify in the figure caption text when a figure is similar but not identical to the original image and is therefore for illustrative purposes only.

See above

6. Please include a copy of Table 2 which you refer to in your text on page 15.

This was actually a typographic error and we meant to refer to Table 1. We have corrected this.

5. Review Comments to the Author

Reviewer #1: I enjoyed reading the manuscript about an interesting topic in earthworm ecology that has been debated since Darwin, who stated that earthworms within a field are patchy distributed without any clear visible differences in the characteristics of the soil.

With a well-designed survey at seven different farms in the UK along an environmental gradient, the authors indeed showed that earthworm density at field scale (within field) was determined by abundance itself and not by any edaphic conditions. At a national scale (between fields and farms), however, earthworm numbers are predicted by density, nitrate levels in the soil, soil temperature and soil moisture. The authors conclude that the patchiness within a field is probably determined by biological factors such as dispersal rate and predation.

This is not new. As the authors also mention in the Introduction (with a list in the Supplementary information), other studies already showed that Darwin was right. Earthworms are indeed patchy distributed without a clear soil characteristics that determines the pattern. However, the authors try to argue that this has consequences for using earthworms as a soil quality indicator. Because of their important role as ‘ecosystem engineers’ and their positive effects on the soil, earthworms are often used as indicators for ecological soil quality, but this is still far from straightforward as they showed with this study. I think this should be highlighted more in this manuscript.

 We appreciate these comments. To better highlight the challenge of using earthworms as indicators of ecological soil quality we have moved our comments on this, which were located in our Conclusions, to a new section that forms the final part of our Discussion. The new section (from Line 423) is titled “Implications of our results for the use of earthworms as indicators of soil health and soil management”.

The structure of the manuscript should therefore be improved. In the analysis and interpretation of the results, the authors should make a clear distinction between national scale, farm scale and field scale. The same is true for total earthworm approach versus the species approach. By doing so, it makes it more easier to interpret the consequences of different sampling designs for earthworms as bio-indicators. Furthermore, the authors should then elaborate more on the consequences of these results in the light of agroecology and how the ecosystems services earthworms provide can be promoted in farmland.

We have modified the manuscript headings to improve clarity. The section in the Results and Discussion that was headed “National and farm level drivers of earthworm distribution.” is now more accurately titled “National level drivers of earthworm distribution” as the data analysis in this section uses the entire dataset as stated in Line 309 at the start of the paragraph on the Bayesian Belief Networks “At a national scale, the Bayesian belief network developed for the entire data set …..” and in Line 339 at the start of the paragraph on species level reponse curves “When species level response curves were considered in the full dataset….”.

We then have a short new section titled “Farm level drivers of earthworm abundance “ (from line 363) in which we present farm level BBNs.

The next section of the Results and Discussion section was headed “Field level spatial patterns of earthworm abundances” and we feel that that heading makes it clear that the section concerns analysis at the field scale, however, to aid clarity we have revised this heading so that it now reads “Field level spatial patterns of earthworm abundances at individual farms”.

“

To better highlight the consequences of our results in the light of agroecology and how the ecosystem services earthworms provide can be promoted in farmland we have moved the text from our conclusions that deal with this issue to our new “Implications of our results for the use of earthworms as indicators of soil health and soil management” section (from Line 423).

We have then revised the remaining text in the Conclusions section so that it is a lot shorter and focuses on our key finding.

I also would recommend the authors to analyse the data based on the ecological group data instead of the species data only, as most studies about earthworms as bio-indicators use the classification epgigeic, endogeic and anecic (e.g. the national survey which is discussed in the Conclusion). It will probably also increase the power as all earthworms (including the juveniles) are included.

We now include tables showing the most influential variables in the Bayesian Belief networks, the sensitivity of the networks and variograms for each farm for each ecotype in the Supplementary information – we have not put this detail in the main paper as it would result in too many figures and tables. However, the key message, which we state in Lines 224-227, 319 – 324 and, 391-393 is that our conclusions hold for each ecotype – we see a similar important of management intensity and environmental variables such as temperature and moisture content for each ecotype in our BBN and a similar level of short range autocorrelation in our variograms. We do see some slight differences in the BBN between the epigeic earthworms vs the anecic and endogeic earthworms and believe this is due to the litter dwelling nature of the epidgeics vs the soil dwelling nature of the endogeics and anecics. However, this does not affect the key message of the paper.

Based on these issues I would recommend major revisions of this manuscript.

Minor issues:

Title: Based on the previous suggestions, I would change the title as other studies already proved that Darwin was right. If the main message is that earthworm abundance is not a good indicator for soil quality, make this clear in the title.

We have changed the title of the manuscript to " The lack of demonstrable control of soil properties on the spatial distribution of earthworms calls into question their ready use as soil quality indicators"

Abstract:

Lines 44-45: From the main text it is not clear how species abundance can be effectively used as ecological indicators.

Lines 45-48: Indicator species are not mentioned in the main text, Remove, repetition of what has been said before.

We have deleted lines 43 – 49 of the abstract and provided alternative wording that we hope better summarises our conclusions “The use of earthworms as soil quality indicators must therefore be carried out with care, ensuring that sufficient samples are taken within field to take account of variability in earthworm populations that is unrelated to soil chemical and physical properties.” 

Introduction:

Lines 55-57: Please add references

References to previous studies are detailed in Table S1 which is referred to later in the Introduction, we now refer to Table S1 here.

Lines 57-58: Please add references

We assume this comment refers to the sentence “The apparent random distribution of earthworms at the field scale has been the object of study over many years.”. Rather than have references at the end of each sentence we feel that the (new) reference to Table S1 at the end of the previous sentence that gives references starting from 1992, together with the subsequent sentence that quotes from Darwin’s 1881 study, provides sufficient references to support this sentence.

Lines 59-61: Although Darwin also ends the paragraph where he mention the patchiness of earthworms, by stating that it is most likely that soil moisture and compaction are important factors determining the distribution of earthworms…

Yes, we go on to mention in the next paragraph different parameters known to influence earthworm distribution. We did not want to quote too much from Darwin’s text but have now added that Darwin suggests that soil moisture and compaction are likely to be important (Lines 54-55). 

Lines 78-79: Please add references, or refer to Table S1 here.

We have moved our reference to Table S1 here, and removed it from the end of the subsequent sentence.

Lines 90-91: Based on these numbers it is most likely that these studies were carried out at field scales, please mention that.

We felt that by giving the dimensions of the plots it would be clear that these were field scale but we have added this information” above studies were carried out at the scale of thousands to tens of thousands of square metres within fields / plots on different” (Line 85-86).

Line 91: “…and often in different climatic zones on different continents.” Than in your study? Why mention this?

We have edited the sentence so that it now reads “Despite being carried out on different land uses and often in different climatic zones on different continents they all report the patchy distribution of earthworms.” (Line 86-88) – the point here is that studies across different land uses, climatic zones and continents all suggest a patchy distribution for earthworms indicating that this is a typical property of earthworm populations and not due to a particular land use, climate etc.

Line 93: Remove the sentence between brackets as you also mention this after this list.

Done

Lines 91-98: Point 1 refers to heterogeneity of the soil and as a result earthworms move to “good” patches (cause-effect). Please elaborate more how predation and parasitism could generate patchiness, and preferably ad references.

We have added a small amount of text here to provide explanations of how predation and parasitism could generate patchiness. As these are standard ecological explanations, and we are not aware of studies that deal specifically with observations of these mechanisms and earthworm populations we cite a popular and well regarded ecology text book (Begon et al., 2006 Ecology: From individuals to ecosystems) as an appropriate reference that contains more information should the reader wish to puruse this point.

New text (lines 93-97) “4) predation resulting in significant reductions in the prey population that can not be balanced by reproduction before the predator moves to a new prey-rich area [18], 5) complex interactions between parasites which cause a reduction in fecundity or an increase in mortality in their hosts and the relative mobilities of the parasites and hosts [18] and, 6) passive dispersal by humans [19] (note here we are not considering the situation where earthworms are invasive species)..”

Lines 105-106: What other land uses do you mean and why would that be relevant for this manuscript?

In particular we mean pasture sites of the sort we study as opposed to arable sites which are the subject of Schmidt and Briones paper – we have now made this clear “However, studies on the effects of varying levels of intensification within other land uses, such as pasture which is the topic of the current study, are still lacking.” Line 105-107.

Lines 114-116: Based on the information in the Introduction, I would hypothesize that the variation in earthworm abundance at the national scale is driven by management intensity and not climate as this is not discussed. Please provide information to the Introduction about climate to justify the hypothesis.

In line 64-65 we already state that “….earthworms are known to respond to a variety of key environmental drivers such as pH, temperature, soil texture, soil moisture and soil density…” (NB we have added soil moisture to this list). We have edited lines 116-118 to make it clearer that we view national scale variations in temperature and moisture content as reflections of climate “….at a national scale the variation in earthworm abundance is driven by climate-related gradients in temperature and rainfall that will be reflected in soil temperature and soil moisture values.”

Methods:

Line 120: All dairy farms? Please mention this

They were all dairy farms, this is now included at line 123

Line 122: Which soil properties were used to select the fields? Soil type?

We have edited this section. Initially farms were selected to give a good coverage of the north-south temperature and east-west rainfall gradients present in the UK at dairy farms where from previous contact with the farmers we knew we would be allowed to sample in four contiguous fields. Contiguous fields were then selected on the basis of farmer advice to give a range of management intensities.

Lines 125-128: What were the average size and age of the fields? Perhaps you can include this information in Table S2.

Table S2 includes details on the minimum length of time that the fields have been used for pasture as stated by the land owners / farmers and we highlight this on line 132. We have added field area to this table. We now give some summary information about field sizes at lines 127 – 128 “Fields ranged in size from 0.97 to 13.34 ha with a mean size of 3.94 ha (full details are given in Table S2).” 

Lines 128-130: What was the management intensity based on? The amount of fertilizer/manure application? Soil disturbance? Traffic intensity? Cattle density? From the Introduction I assume that it is disturbance, but how does this look like on a farm? Were the field with highest intensity the youngest?

Our definition of management intensity was deliberately left undefined to allow the farmers to apply their expert knowledge of their farms without constraining them to specific definitions, e.g. amount of fertiliser application or traffic intensity. This is because the farms will have been managed differently and thus a single quantitative metric for management intensity will not necessarily be transferable across farms. We have deliberately used BBN to analyse our data to take into account the qualitative nature of our definition. We have measured a range of quantitative metrics that are used in our analysis to investigate controls on earthworm populations, such as soil density and nitrate levels that will relate to aspects of management intensity and simply asked the farmers to select fields of differing intensity and to rank them so that we sampled over a range of management intensity levels and were able to qualitatively compare earthworm abundance across the gradient (see Fig. 2b). The only analyses in which we use the intensity measure is where we qualitatively observe that for each field earthworm abundance did not increase with management intensity (Lines 303-304) and when we produce BBN for each individual farm (e.g. Table 2) where the analysis is within farm rather than between farm. There is no relationship between the length of time that the fields have been used as pasture and their level of intensification – this data is given in Table S2 in the supporting information which is referenced on Line 132. We have edited Lines 135-139 to emphasise that our definition of “management intensity” is based on expert judgement rather than a specific metric.

As this classification is qualitative and based on discussion with the farmers, perhaps you could make it more quantitative by using nitrate levels or bulk density as a measure of management intensity?

The overall measure of management intensity, although qualitative and expert based, is valuable to use for two reasons; i) in any farm management or environmental payments scheme, these will often be based on simple, aggregate measures or interventions (i.e. what the farmer does or does not do) rather than more costly soil measures, so it is valuable to see if they on aggregate relate to earthworm population distributions as ecological indicators of soil quality; and ii) as we include these quantitative measures already in our BBN’s, this allows us to explore whether there is more to management intensity than the obvious measurable soil properties to which the earth populations may be responding. 

Line 130-131: Was this done before or after the selection of the fields?

Farmers / land managers were asked to identify four fields of differing management intensity for sampling. Thus the ranking of the fields was an integral part of the selection of the fields

Lines 143-146: What about freezing conditions during fieldwork? Earthworms will also die or aestivate under freezing conditions which were more likely to occur during the fieldwork period.

We have added further text at lines 157-158 further justifying our choice of sampling season. “Work conducted for several years in the New Forest has shown that this is the optimal period to sample earthworms. During the sampling campaign none of the fields froze and all had active worms throughout the whole of the sampling period. There may have been overnight frosts towards the end of the sampling period but if there were they did not noticeably affect earthworm activity. “

Lines 150-152: How were the number of samples per field determined?

The total number of samples, across all fields per farm approximated 100 samples to be able to develop robust estimations of the variograms; the 100 samples per farms distributed over the fields resulted in an average of 20 per field. 

Lines 160-161: Why?

We now explain why we did this (Lines 175-177) “Pits falling within 5 m of the field boundary were reassigned a new random position, as field margins are known to have a different invertebrate assemblage from those in the middle of the field [29]. “

Lines 172-173: What is the rationale behind measuring vegetation cover? The unit is % covered soil?

Vegetation cover was measured as variation in plant species has been shown to impact on earthworm distributions (see for example Babel et al (1992) Soil Biology and Biochemistry 24 1477-1481). We have added vegetation to the list of factors affecting earthworm abundance in Line 69 in the Introduction and edited Line 188 to confirm that the units is % of covered soil.

Lines 179-180: Already mentioned in lines 161-163

We have deleted this text.

Lines 207-210: Which soil properties were included? How were field management and landscape attributes included (units)? 

The soil properties included are those described in section on site, soil sampling and analysis. Further explanatory variables were obtained from the LandIS database (Soil Survey of England and Wales - http://www.landis.org.uk), which was intersected with the sample locations and from a digital terrain model (represented in Figure 1), which resulted in a range of landscape variables; ,Elevation, Aspect, Slope, Elev, Soil association, soil unit, soil description, dominant geology, dominate soil series in the association, associated soil series in the associate, historical crop/land-use in LandIs. 

BBN’s are capable of analysing a mixture continuous and categorical data; the categorical classes as individual ‘states’ in each node; the software used bins for the continuous data as this is more computationally efficient; the bins become ‘states’ of the continuous data nodes.

Line 219-221: How was the response variable entered? What were the explanatory variables? Did you use a general linear model (as mentioned in the results section) or a generalized linear model (as mentioned here)? If so, what was the random factor?

We have added text (Lines 250-257) to clarify this.

Results and Discussion:

Lines 297-298: With temp 5 cm most important and %carbon least important?

There is no rank order in this. 

Lines 348-352: The main conclusion is that the distribution within fields is likely to be determined by biotic conditions, however, the argumentation for this is very thin, please elaborate more on this and add more references to your conclusions. How could reproduction rate explain the patterns? Or dispersal? Or predation? And is this equal among species or ecological groups?

We refer to these possible mechanisms in the introduction – see Reviewer comment on lines 91-98 above and have added lines 407-409 to state that these are just hypotheses of possible biotic factors that may influence the response of worms. We didn’t attempt to measure competition or predation or parasitism etc, so further data is needed rather than more statistical analyses and thus we do not think it appropriate to expand on differences between species / ecological groups. Our other edits to the manuscript indicate that our conclusions on the distribution of earthworms and their lack of correlation with soil physical and chemical properties apply to all ecotypes.

Conclusions:

Lines 388-391: What were the threshold qualitative vales? And were these also for pasture land? As Stroud also looked in arable land. And what about other studies that use earthworms as indicators?

This text has been moved into our new section “Implications of our results for the use of earthworms as indicators of soil health and soil management”. On Line XXX we clearly state that this was a method for arable fields. The threshold for determining good vs poor soil quality that Stroud defined was for 66% of sampled pits to contain 16 or more earthworms. We now state this at Line 440-442. Some of the other soil quality schemes we cite on Line 434 suggest numbers of earthworms that represent good or poor quality soil e.g. [7], [8]; whilst others do not, e.g. [6]; in some schemes absolute numbers of earthworms representing different levels of soil quality are deliberately not given, see discussion in [9] for example. Similarly some schemes suggest numbers of pits to be sampled per field (e.g. Stroud suggests 5 may be appropriate, [7] and [8] suggest sampling 3 or 4 sites per field) whilst others do not. We do not want our paper to become a critical review of soil quality assessment schemes, rather the purpose of our study was to investigate controls on earthworm distribution and think about its implications. For this reason we highlight the Stroud scheme [11] as an example of how the patchy nature of earthworm distribution necessitates sampling at multiple pits to obtain useful information for application of soil quality assessment schemes but would rather not go on to assess each soil quality scheme in the literature. However, in lines 434 we now also highlight that [7] and [8] recommend the use of 3 or 4 sample sites per field which our results suggest would give unreliable estimates of earthworm numbers.

Lines 405-407: It would be nice if you could argue a bit further how this would look like.

We have rewritten our conclusions (Lines 465-483).

Reviewer #2: This study is an interesting and statistically rigorous examination of spatial variability in earthworm communities across multiple scales. Similar studies have generally looked at a single field in detail or have looked at more sites but only very coarsely, and thus I think it is a very useful contribution to knowledge in this area. I have a number of minor comments:

Line 68 – specify that it is healthy “agricultural” soil that they are suggested to be good indicators of

Done

Table S1 – since there are other studies that have examined earthworm distributions in forests that are not mentioned at all, it would be good to include in the title that these are studies that investigated the spatial distribution of earthworms in “agricultural/savannah” habitats. (I agree it makes sense to only list studies in these open habitats though)

Done

In line 92 - 98, there is a list of how patchiness in earthworm populations can be generated. What about cases where earthworms are spreading in a field for the first time either because disturbance removed them from the area, or because they are invading? Then introduction by humans or passive dispersal by humans or natural vectors could also lead to patchiness (see discussion in Eijsackers 2011 Applied Soil Ecology). These potential sources of patchiness do not seem like they are quite encompassed in any of the items in the list.

We now reference Eijsackers (2011) in the text at lines 97 and state that passive dispersal may also account for patchiness (lines 97-98).

Line 114-116 – There is some discussion about how management intensity affects earthworm abundance earlier in the introduction, but not much mention of climate (only in a list with other drivers on line 72), but then the hypothesis here is that climate will drive earthworm abundance at a national scale. I think this is a reasonable hypothesis but it doesn’t seem to follow clearly from the introduction – can you add some explanation (could cite another paper or two (e.g., Phillips et al. 2019 Science)) to provide more background for this hypothesis?

We have added in to line 76 that soil moisture is also an important driver – we believe that given that temperature and moisture content are identified as drivers of earthworm abundance this is a good indication that climate will play a role in their abundance. However we have added two references to line 68 that specific mention climate gradients “to climatic gradients more broadly which, for example combine effects of temperature and rainfall, e.g. [13], [14] and”

Line 155 – Somehow it is still slightly hard to picture how the pits were positioned (was the field divided into a grid and then the random numbers were used on that, or something?). Please clarify this explanation if possible. 

The pits were distributed randomly across a field ensuring that the distances between the pits varied sufficiently to allow for an effective estimation of the variograms and coverage across the field. We have added a small amount of text at Line 170-174.

Line 176 For the soil sampling, what depth was the soil collected from? It sounds like it was just collected from the bottom of the soil pit, but was it actually a 10 cm depth core? What volume of soil was collected for the bulk density measurement?

We have now added text at lines 193-197 to clarify this. 

Statistical analysis – I haven’t ever run a BBN model but that analysis seems well-explained. Why did you switch over to AIC/p-values for species response models, versus continuing with a Bayesian approach? 

We used Bayesian methods when we were dealing with complex mixtures of expert knowledge, categorical and continuous data. The species response curves were more straightforward and could be analysed using standard approaches We have added this information at lines 217-221. 

Although the explanation of the variograms/cross variograms is pretty standard I think, could you add a reference here?

We have added a reference [42] 

Line 390 – can you provide any details on how the power analysis was done?

The power analysis was done as we describe it on lines 440-444, using the observed variance in earthworm populations, set to determine a difference of > 66% of pits containing 16 earthworms or more, threshold set by Stroud (2019)

Figure 5 – this figure seems somewhat disorganized e.g., is the panel that is between a and b part of a) or b)? Placing the variograms in the same position within each panel and adding labels to some of the maps might help. The file names of the layers such as “[DST.txt_Features].[TOTall]” could be deleted or changed so that they are more informative for readers as well.

We have revised this figure.

---

## [Decision Letter · Decision Letter 1]

18 Aug 2021

Earthworm distributions are not driven by measurable soil properties. Do they really indicate soil quality?

PONE-D-20-33339R1

Dear Dr. Eggleton,

We’re pleased to inform you that your manuscript has been judged scientifically suitable for publication and will be formally accepted for publication once it meets all outstanding technical requirements.

Please make sure to proofread the manuscript for tyopgraphic errors or missing connector words prior to final submission, especially in the sections of revised and added text.

Kind regards,

Manu E. Saunders

Academic Editor

PLOS ONE

Additional Editor Comments (optional):

Reviewers' comments:

Reviewer's Responses to Questions

**Comments to the Author**

1. If the authors have adequately addressed your comments raised in a previous round of review and you feel that this manuscript is now acceptable for publication, you may indicate that here to bypass the “Comments to the Author” section, enter your conflict of interest statement in the “Confidential to Editor” section, and submit your "Accept" recommendation.

Reviewer #2: All comments have been addressed

2. Is the manuscript technically sound, and do the data support the conclusions?

Reviewer #2: Yes

3. Has the statistical analysis been performed appropriately and rigorously? 

Reviewer #2: Yes

4. Have the authors made all data underlying the findings in their manuscript fully available?

Reviewer #2: Yes

5. Is the manuscript presented in an intelligible fashion and written in standard English?

Reviewer #2: Yes

6. Review Comments to the Author

Reviewer #2: All of my previous comments were addressed adequately. One thing that I think should be corrected is that in line 116, it says that the study tested whether spatial variation was random at the field level. However, what it tested was actually whether soil/climate variables affected distributions, not whether distributions were random, so this should be changed. Also, in line 298 there is a typo in Aporrectodea caliginosa and in line 257, it should say "mixed models" not "model".

7. PLOS authors have the option to publish the peer review history of their article (what does this mean?). If published, this will include your full peer review and any attached files.

Reviewer #2: No

---

## [Editor Report · Acceptance letter]

20 Aug 2021

PONE-D-20-33339R1 

Earthworm distributions are not driven by measurable soil properties. Do they really indicate soil quality? 

Dear Dr. Eggleton:

I'm pleased to inform you that your manuscript has been deemed suitable for publication in PLOS ONE. Congratulations! Your manuscript is now with our production department. 

Kind regards, 

on behalf of

Dr. Manu E. Saunders 

Academic Editor

PLOS ONE